# Attitudes and Perceptions of Local Communities towards Nile Crocodiles (*Crocodylus niloticus*) in the Sudd Wetlands, South Sudan

**DOI:** 10.3390/ani14121819

**Published:** 2024-06-18

**Authors:** John Sebit Benansio, Gift Simon Damaya, Stephan M. Funk, Julia E. Fa, Massimiliano Di Vittorio, Daniele Dendi, Luca Luiselli

**Affiliations:** 1AERD—Alliance for Environment and Rural Development, El Hikma Medical Centre Street, Gudele West, Block II., Juba P.O. Box 445, South Sudan; sebitbenansio@gmail.com; 2Department of Wildlife Science, University of Juba, Juba P.O. Box 82, South Sudan; gftsimon@yahoo.co.uk; 3NatureHeritage, St. Lawrence, Jersey, Channel Island JE2 3NG, UK; 4Department of Natural Sciences, School of Science and the Environment, Manchester Metropolitan University, Manchester M1 5GD, UK; jfa949@gmail.com; 5Center for International Forestry Research (CIFOR), CIFOR Headquarters, Bogor 16115, Indonesia; 6Ecologia Applicata Italia s.r.l., Termini Imerese, 90018 Palermo, Italy; 7Institute for Development Ecology Conservation and Cooperation, Via G. Tomasi di Lampedusa 33, 00144 Rome, Italy; d.dendi@ideccngo.org (D.D.); l.luiselli@ideccngo.org (L.L.); 8Department of Applied and Environmental Biology, Rivers State University of Science and Technology, Port Harcourt P.M.B. 5080, Nigeria; 9Département de Zoologie et Biologie Animale, Faculté des Sciences, Université de Lomé, 101 B.P., Lomé 1515, Togo

**Keywords:** *Crocodylus niloticus*, questionnaires, fisherfolk, attitudes and perceptions towards conservation, Sudd wetlands, East Africa

## Abstract

**Simple Summary:**

South Sudan’s recent recovery from armed conflict presents an opportunity to address critical conservation issues affecting the country’s biodiversity. The protection of the vast Sudd wetlands is vital for the conservation of many different species and habitats and to ensure the continuity and improvement of the lives of human communities living in it. Animal–human conflict, particularly from crocodiles, poses a significant threat to the adequate protection of the Sudd wetlands. Crocodile attacks have resulted in mortality rates ranging from 50% to 100%. To mitigate these conflicts, changing human behaviour through environmental education is key. This can also improve attitudes towards biodiversity conservation, aligning future development with conservation needs. We conducted interviews with fishers to understand resident people’s perception of crocodiles. Crocodiles are seen as a threat because they restrict movement along water bodies, attack livestock and humans, and damage fishing equipment. Attitudes are complex, nuanced, and sometimes polarised within communities. They are feared and hated but also valued for their meat and skin. Some interviewees believe that consuming crocodile meat can improve longevity, sexual potency, and protect against witchcraft. While there is a consensus on the need to destroy crocodile breeding habitats, there is also support for establishing protected areas in the Sudd wetlands. Crocodile sanctuaries would help reduce illegal hunting and protect the species, especially with the growing human population and economic development after the civil war. The nuanced attitudes revealed in certain questions provide a valuable foundation for raising awareness and designing more targeted promotional campaigns.

**Abstract:**

Conflicts between human populations and Nile crocodiles are widespread with crocodiles posing significant threats to fisherfolk and riverine communities across r-Saharan Africa. Hundreds of deadly attacks take place annually, and mortality rates may range from 50% to 100%. Attitudes and perceptions towards crocodiles were studied using structured questionnaires among fisherfolk along the River Nile and the Sudd wetlands in South Sudan. Local communities used crocodiles for their meat and skin/leather trades. The meat is regarded to enhance longevity, sexual potency, and protection against witchcraft. Crocodiles are perceived as a main threat to lives and livelihoods as they restrict people’s freedom of movement along water bodies, attack livestock and humans, and devastate fishing equipment. To assess whether responses were influenced by the intensity of crocodile threats, published data on fatal crocodile attacks on humans and livestock were analysed using Generalised Linear Models (GLMs). This analysis indicated a direct link between the number of crocodile attacks and human attitudes. Crocodiles were generally feared and hated, and there was the agreement of the need to destroy breeding habitats. However, some attitudes were complex and nuanced as highlighted by the agreement of local communities on the need to destroy Nile Crocodile breeding habitats on the one hand and the need to establish crocodile sanctuaries as the the preferred strategy to mitigate risks and conflict on the other hand. There is a need for the creation of a crocodile sanctuary in the Sudd wetlands to minimise the risks of illegal hunting and to buffer the increasing pressure on crocodiles due to human population growth and economic upturn after the civil war.

## 1. Introduction

Crocodiles are apex predators in freshwater ecosystems throughout tropical regions. Conflicts between crocodiles and human populations date back to the Plio-Pleistocene when these reptiles preyed on earlier hominids in Africa [1]. Nile crocodiles (*Crocodylus niloticus*) are among the main threats to fisherfolk and riverine communities in Africa [2,3,4,5,6]. It is estimated that hundreds of deadly attacks attributable to this species occur yearly in Sub-Saharan Africa including South Sudan, with a mortality rate ranging between 50% and 100% [7,8]. Nile crocodiles not only pose a direct threat to people and livestock, but they can also indirectly affect the quality of life of people in more remote locations and economically impoverished areas [9]. On the other hand, Nile crocodiles are also valued prey for humans because of their meat and skins [10]. Although considered a Least Concern species by IUCN [11], this species is exposed to local declines [12,13] and, therefore, needs continued monitoring, including using innovative survey techniques [14].

Thus, the relationships between Nile crocodiles and human populations are often complicated and need careful examination and monitoring even at the local scale, especially in the scientifically poorly explored, thus data-deficient regions and the poorest areas of the African continent. South Sudan in East Africa is a prime location to study crocodile–human conflicts as it is a scientifically poorly explored region because of the civil war that has run throughout the country for more than 20 years. Considering the importance of the vast South Sudanese wetlands for both human food security through artisanal fishing and as a prime habitat for Nile crocodiles, the potential for crocodile–human conflicts is very high given that (i) the density of people is relatively high around the Nile, (ii) most of the economic activities are concentrated along the Nile, (iii) the country is a low-income food-deficit country with a per capita GDP limited to USD 1570 [15], whereby 21% of the population was at level four emergency under the Integrated Food Security Phase Classification in 2018 [16], and (iv) expansive areas are characterised by riverine and freshwater marshlands (for instance, the Nile river and the Sudd wetlands), thus providing potentially excellent habitats to Nile crocodiles [17,18,19]. A previous study documented a high frequency of attacks by Nile crocodiles on both humans and livestock in the Sudd wetlands, with substantial mortality for fisherfolk and riverine communities [8].

Given the severity of conflicts between humans and crocodiles, changing human behaviour through environmental education could mitigate these conflicts. Such behavioural changes could also positively impact attitudes towards biodiversity conservation more broadly. This, in turn, could help ensure that future economic development and human population growth are more compatible with conservation needs. Our objective was to lay the groundwork for this effort by gaining a deeper understanding of human perceptions and attitudes towards crocodiles. In this paper, we investigated crocodile–human conflicts in South Sudan through a standardised interview survey on fisherfolk and other people living in the riverine area to respond to these key questions:-How do Nile crocodiles affect the lives and livelihoods of local communities?-What are the attitudes of local communities towards Nile crocodiles?-What strategies are used by local communities to minimise/mitigate the risks of Nile crocodiles’ attacks on human and livestock?-Can the knowledge of the attitudes of local communities be used for applying any management and conservation strategies for Nile crocodiles in South Sudan?

## 2. Materials and Methods

South Sudan lies within the tropical zone between latitude 3.5° and 12° north and longitude 25° and 36° east, and it occupies an estimated area of 633,906 km^2^ (in 2018, [20]). It is a landlocked country in the Nile River Basin in east–central Africa. The regional climate is tropical with a wet season in April–October (with an average of 100 mm rainfall per month) and a dry season between November and March (5–35 mm per month). During the dry season, the maximum temperature of 38 °C is typical in February. 

Poverty is prevalent, with approximately 80% of the population living on less than USD 1 per day [21,22]. Nearly 80% of the population relies on smallholder agriculture, farming, and fishing. Fish is crucial for both the food security and livelihoods of communities in and around the Sudd wetlands, thus creating potential human–crocodile conflicts [8]. 

Designated as a Ramsar site in 2006, the expansive Sudd wetlands rank among the largest and species-rich wetland areas globally [23]. Comprising lakes, marshes, and extensive floodplains, the Sudd is renowned for its biodiversity, serving as a critical habitat for numerous endangered species such as the Nile lechwe and the shoebill stork. Its biodiverse aquatic habitats provide essential habitats for fish, offering ideal grounds for their spawning, rearing, and feeding. These habitats remain largely untouched by industrial development, preserving their ecological integrity [24]. 

The Sudd wetlands are also vital for local livelihoods, providing ecosystem services and supporting water resources. However, they also present important challenges to humans living in the area, one of which are conflicts between humans and crocodiles, as the wetlands are a prime habitat for Nile crocodiles. In the south, the wetlands are bordered by the Badingilo National Park (also spelled Bandingilo National Park), which is one of the most important migration areas for wildlife in Africa, second only to the Serengeti Mara ecosystem. The park serves as a critical corridor for the migration of various antelope species, including white-eared kob and tiang, which undertake one of the largest terrestrial migrations in the world.

We studied 21 different villages (Appendix A) and their 85 associated fishing camps from the following five administrative areas of the Southern Zone of Sudd wetlands, mainly situated in Central Equatoria State (Figure 1): (1) Terekeka (N050 27.1555″ and E0310 45.268) and (2) Northern Terekeka (N050 38.829″ and E0310 43.108″) on the western bank of the Nile; (3) Mangalla on the eastern bank (N050 11.5350″ and E0310 46.164″); (4) Gemeiza on the western bank (N050 44.2738″ and E0310 47.1021″); and (5) Tombek (N050 47.369″ and E0310 42.285″). All these locations are situated surrounding the western and eastern corridors of the Badingilo National Park and include the large swamps 40 km east of Mangalla Payam in Central Equatoria State.

The Badingilo National Park (about 8400 km^2^) is characterised by grassland and woodland savanna and is known to be the earth’s second-largest hotspot for ungulate annual migration after the Serengeti. These abundant populations of multiple species of ungulates support large crocodile populations who prey on them [8,25]. 

The diversity of natural resources has contributed to the expansion of human settlements near the Badingilo National Park, whereby the main economic activities fishing, livestock grazing, illegal hunting/poaching of wildlife, charcoal production, collection of reeds and other building materials, collection of fuel wood, and the production of crafts are important aspects in the rural economy. Vegetation and sandy areas along the bank of the Nile provide suitable nesting sites for the Nile crocodiles [8]. The total human population of the study areas was 47,718 in 2018 [26]. All study sites are characterised by near proximity of humans and Nile crocodiles (Figure 2).

**Figure 1 animals-14-01819-f001:**
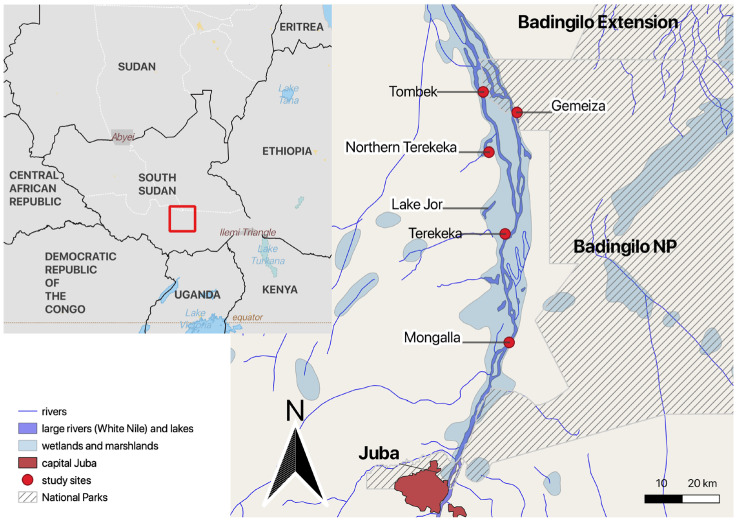
Study area. The map was created using QGIS version 3.20.2-Odense (qgis.org) from public domain map datasets from Open Street Map (www.openstreetmap.org, accessed on 24 January 2024), diva-gis (diva-gis.org), Humanitarian Data Exchange, HDX (data.humdata.org), and UNEP-WCMC [27] for the boundaries of the Badingilo National Park.

The interview survey was carried out from 2018 to 2020 by a team of three researchers. We randomly selected 28 of the 85 fishing camps for interviews. The random selection was across the five administrative areas without a proportional representation of the administrative areas. Male participants were randomly chosen from fisherfolk, farmers, and shepherds who lived in the fishing camps and villages around and inside the Badingilo National Park. Only males live in the fishing camps, according to local culture, since women visit their partners to cultivate vegetables and crops around the camps during the dry season when water levels are low. In our surveys, women were under-represented because they required their male partner’s permission to speak, which was rarely granted. Potential participants were randomly approached when encountered when interviewers visited the fishing camps. A total of 378 local community members were interviewed, of which 21 were women and 357 were men. Each interview was conducted individually and lasted between 20 and 25 min. Only adults age 19 or above were questioned; the age distributions were 19–29 years old, *N* = 103; 30–49 years old, *N* = 189; 50–69, *N* = 76; and 70 years and above, *N* = 10. Interviewees were given the option to decline the complete interview or single questions. After each interview, the name of the interviewee was requested (optional) and a reference number recorded to avoid replication. Five main questions were asked: -How do Nile crocodiles affect the lives and livelihoods of your local community regarding (Q1.1) restricting freedom of movement at the river bank, (Q1.2) attacking livestock, (Q1.3) destroying fishing equipment, and (Q1.4) attacking community members?-What is the attitude of local communities in your area towards Nile crocodiles regarding (Q2.1) danger, (Q2.2) honour, (Q2.3) hate, (Q2.4) fear, and (Q2.5) enemy?-How are Nile crocodiles being used by the local community in your village regarding (Q3.1) local meat trade, (Q3.2) illegal farming practices, (Q3.3) skin/leather trade, (Q3.4) ornamental purposes, and (Q3.5) religious purposes?-What are the perceptions of local communities towards eating Nile crocodile meat at your village regarding (Q4.1) favouring longevity, (Q4.2) enhancing sexual performance, (Q4.3) for anti-witchcraft, (Q4.4) for medicine purposes, and (Q4.5) boosting business and good luck?-What strategies are preferred by your local community to minimise/mitigate the risks of Nile crocodile attacks on humans and livestock regarding (Q5.1) creating a sanctuary/protected area for crocodiles with no access to people (while allowing free hunting outside it), (Q5.2) reducing the livestock activities along the rivers, (Q5.3) destroying the crocodile habitat to make them leave the area, (Q5.4) emigration of the whole community to safer places with no crocodiles around, and (Q5.5) promoting the hunting of crocodiles?

These questions were asked in the local language and verbally phrased such that answers could be given on a Likert scale: (a) strongly agree, (b) agree, (c) neutral, (d) disagree, or (e) strongly disagree (thereafter, “agreement options”). For example, Q2.1 was formulated as “Do you perceive that crocodiles pose a danger to you?” The information provided by the interviewees was augmented through group discussion in each location to validate the individual interviews on a group level. Following these questions, we provided the interviewees with an opportunity to express their attitudes towards crocodiles and biodiversity conservation, aiming to gather general background information.

After completing the interviews, in-depth telephone interviews were held with six senior government officials (two from the HQ Office in Juba, two from the State Department of Wildlife, and two from the Terekeka County Authority). Qualitative data were gathered from key informants including thirteen village chiefs or community leaders and sixteen heads of fishing camps to understand their views towards the conservation of Nile crocodiles. All this information guided us in interpreting the collected quantitative data.

All work was undertaken under a research permit from the Ministry of Internal Affairs, National Government, Juba, South Sudan. Because of the COVID-19 pandemic, the research team members used the relevant safety measures defined by the Ministry of Health of South Sudan when making the face-to-face interviews, including social distance measures and the use of face masks throughout the research period.

The various frequencies of answers were analysed by contingency tables χ^2^ tests. To evaluate whether answers were influenced by the intensity of the threats due to crocodiles, we used the dataset provided by Benansio et al. [25] who reported, for the period 2018 to 2020, a total of 23 fatal crocodile attacks on humans and 355 attacks on livestock, of which 166 were killed. Stratified by district, these data were used to assess their effect on answers. 

Generalised Linear Models, GLMs [28,29], were used to test the relationship between questionnaire answers (from Q1.1 to Q5.5) and crocodile attacks on humans and livestock in the five surveyed village districts. GLMs can handle ordinal response variables including Likert scale responses by using appropriate link functions. GLMs do not assume that the response variable has a normal distribution, which is often violated in Likert scale data. The values of the estimation parameters in the Generalised Linear Model are obtained by maximum likelihood (ML) estimation through iterative computational procedures. Tests for the significance of the effects in the model can be performed via the Wald statistic. Detailed descriptions of these tests can be found in McCullagh and Nelder [29]. In the models, answers were used as dependent variables, and the numbers of crocodile attacks on humans and livestock were used as predictors. All models were computed with the all-effects procedure, the identity link function, and a normal distribution of errors [30]. The estimates of the models indicated whether the probability of a given answer increased/decreased with the increasing number of crocodile attacks. Attacks on humans and livestock were considered separately as predictors, but the number of attacks on all types of livestock (cattle, goats, and sheep; see Benansio et al., [8]) was cumulated for model evaluation. 

Software STATISTICA 13.0 was used to perform all the analyses, with alpha set at 5%. Nonparametric tests were used when the variables were not normally distributed.

## 3. Results

The distributions of the answers to the questions are presented in Figure 3 and Table A1, Table A2, Table A3, Table A4 and Table A5 in Appendix B. There were no frequency differences among study areas (in all pairwise comparisons, *p* > 0.15 at χ^2^ tests), and we pooled the samples for further analyses.

### 3.1. How Do Nile Crocodiles Affect the Lives and Livelihoods of Your Local Community?

There were statistically significant differences between the frequencies of the five agreement options, with the highest percentage of respondents selecting the option “strongly agree” for each of the four types of potential factors (Q1.1: χ^2^ = 54.9, df = 4, *p* < 0.0001; Q1.2: χ^2^ = 101.2, df = 4, *p* < 0.0001; Q1.3: χ^2^ = 59.3, df = 4, *p* < 0.0001; Q1.4: χ^2^ = 105.8, df = 4, *p* < 0.0001; Figure 3a). Thus, the distribution of the answers indicated that crocodiles are perceived as negatively affecting the interviewee’s lives. Fisherfolk in the Southern Zone of Sudd wetlands annually lose an estimated number of more than 350 fishing nets as a result of Nile crocodile damage. Most of the fishing net damage was reported for the peak fishing season especially when most fish species were migrating for feeding and breeding grounds in the floodplain areas. The damage of fishing nets, as well as the attacks on humans and livestock by crocodiles, have increased the human–crocodile conflict in the Sudd wetlands. The research team observed most of the human settlements are close to the edge of the river bank, and at the same time the local communities are raising their livestock in the critical zone of Nile crocodiles. For instance, focus group discussions highlighted that Nile crocodiles attack livestock at night mainly in the dry season when there are no more fish in the floodplain. The in-depth interview highlighted that there is a high incidence of juvenile crocodile by-catch during the peak fishing season. However, the fisherfolk do not report such cases to the concerned authorities due to fear that they will be arrested, and their fishing nets would be confiscated. Instead, fisherfolk communities used the juvenile crocodile by-catch to “compensate” the losses of fishing nets. They use the juvenile crocodile by-catch as food, sell live juvenile crocodiles to individuals interested, as well as the dead juvenile crocodiles for ornamental purposes.

### 3.2. What Is the Attitude of Local Communities in Your Area towards Nile Crocodiles?

The great majority of the respondents “strongly agreed” (Figure 3b) with fear (χ^2^ = 71.8, df = 4, *p* < 0.0001), hate (χ^2^ = 65, df = 4, *p* < 0.0001), danger (χ^2^ = 86.5, df = 4, *p* < 0.0001), and enemy (χ^2^ = 80.2, df = 4, *p* < 0.0001), whereas they strongly disagreed that crocodiles may convey any “sense of honour” to their communities (χ^2^ = 54.1, df = 4, *p* < 0.0001). The overall result was a largely negative attitude. The relationship between Nile crocodiles and local communities is perceived as negative due to the fear that these reptiles attack both humans and livestock. Despite these negative relationships, the key informant interviews cited that some families and clans believe that Nile crocodiles have cultural and ritual/spiritual purposes. For instance, if they cannot perform ritual functions, the pastoralist communities will not cross with their cattle to other sections of the river bank because of fear of crocodile attacks on humans and livestock. In such cases, the pastoralist communities have to provide fresh milk and offer a bull that can be slaughtered as a sacrifice for the ritual function. Once the ritual function is performed, the pastoralists will start crossing with the cattle to the other site of the river bank. The in-depth interviews highlighted that there are fourteen (14) cultural and spiritual sites associated with the Sudd wetlands located in Terekeka, Mangalla, and Gemeiza County of Central Equatoria State. Terekeka has eight (8) cultural and ritual sites, Mangalla with five (5), and Gemeiza with two (2). Out of the fourteen (14) cultural and ritual sites associated with magic and taboo in the Southern Zone of Sudd wetlands, ten (10) of these cultural and spiritual sites are strongly associated with beliefs in crocodiles as their ancestor or god. Data generated from the in-depth interviews show that there are more than thirty-four (34) cultural and spiritual sites along the Sudd wetlands covering Central Equatoria, Jonglei, Lakes, Unity, and Upper Nile State.

### 3.3. How Are Nile Crocodiles Being Used by the Local Communities in Your Village?

Most respondents “strongly agreed” (Figure 3c) that crocodiles are used in the local meat trade (χ^2^ = 77.6, df = 4, *p* < 0.0001) and skin/leather trade (χ^2^ = 35.7, df = 4, *p* < 0.0001), “strongly agreed” or “agreed” for farming practices (χ^2^ = 32.4, df = 4, *p* < 0.0001), whereas they significantly selected the “strongly disagree” option for ornamental purposes (χ^2^ = 14.9, df = 4, *p* < 0.01) and for religious purposes (χ^2^ = 51.4, df = 4, *p* < 0.0001). Indeed, there is a high demand for Nile crocodile meat at the level of households, restaurants, bars, hotels, and lodges, especially in Juba, the capital city of South Sudan. The high demand for crocodile meat will likely contribute directly or indirectly to illegal poaching to meet the demand. The Ministry of Wildlife Conservation and Tourism approved permits for the hunting of Nile crocodiles based on requests from individual and government officials. It seems the number of Nile crocodiles hunted could not meet the high demand for Nile crocodile meat both in rural and urban centres. Hence, the ongoing illegal hunting of Nile crocodiles in the Southern Zone of Sudd wetlands aims to meet the demands of Nile crocodile meat in both rural and urban centres. Therefore, promoting Nile crocodiles farming could be the best alternative to reduce the pressure of hunting Nile crocodiles in their natural habitats. There is ongoing illegal trading of Nile crocodile skin through smuggling to the neighbouring countries. Indeed, before the conflict in Sudan arose, Nile crocodile skin was smuggled from Sudd wetland via river transport up to the Sudanese capital Khartoum. 

### 3.4. What Are the Perceptions of Local Communities towards Eating Nile Crocodile Meat in Your Village?

Most answers indicated that crocodile meat was considered a powerful remedy for enhancing sexual prowess (Figure 3d; “strongly agree” and “agree” options: χ^2^ = 84.7, df = 4, *p* < 0.0001), to enhance longevity (χ^2^ = 38.6, df = 4, *p* < 0.0001), and as anti-witchcraft (χ^2^ = 67.8, df = 4, *p* < 0.0001), whereas no other reasons were suggested to explain the consumption of this type of meat (*p* > 0.05). The local communities perceived that the meat of crocodiles is excellent for human health. Furthermore, there is a strong traditional belief that crocodiles are among the most resilient reptile. Therefore, eating crocodile meat can prolong life and enhance sexual performance. However, there is no scientific proof and evidence for these claims. The focus group discussion cited that it is impossible to find dead crocodiles in the Sudd wetlands areas because the ecosystem is healthy and there is plenty of food except for those killed by bullets or spears. Local fisherfolk use crocodile abundance to estimate ecosystem health and fish abundance. The focus group discussion stressed that pregnant women are not allowed to eat crocodile meat because of fears of resulting in miscarriage. 

### 3.5. What Strategies Are Used by Your Local Community to Minimise/Mitigate the Risks of Nile Crocodile Attacks on Humans and Livestock?

There were statistically significant differences between the frequencies of the five agreement options across strategies (Q5.1: χ^2^ = 41.16, df = 4, *p* < 0.0001; Q5.2: χ^2^ = 72.2, df = 4, *p* < 0.0001; Q5.3: χ^2^ = 12.89, df = 4, *p* < 0.05; Q5.4: χ^2^ = 12.02, df = 4, *p* < 0.05; Q5.5: χ^2^ = 22.5, df = 4, *p* < 0.0001). Promoting crocodile killing or destruction of their habitats was not considered a crucial strategy to reduce the risks of crocodile attacks by the majority of people (Figure 3e). Instead, the creation of a protected area to conserve crocodiles without allowing people to enter was positively considered (“strongly agree” answer) by the majority of the respondents (Figure 2e). The local communities have used various strategies to mitigate/reduce the risk of crocodile attacks on humans and livestock. However, some of these strategies are positive and others are negative to the ecosystem. The positive strategy suggested by local communities is the creation of a crocodile sanctuary in the Sudd wetland hotspot to minimise the risks of attacks on humans and livestock. The local communities cited that awareness and outreach programs on crocodile–human conflicts are suitable to minimise/reduce the risk of crocodile attacks. The key informant interviews stressed that there is a lack of awareness and outreach programs. For instance, the people tend to build their houses and set up cattle camps less than 10 m from the river’s edge, thus exposing themselves to crocodile attacks. Most children are vulnerable to crocodiles when they swim and when they play in sandy areas suspected to be the breeding grounds of Nile crocodiles. Women can be victims of crocodiles while washing clothes and fetching water. Therefore, providing basic services such as building schools and community health centres, establishing a borehole and water pumps, and creating water points for livestock are the best strategies to minimise/reduce the risks of crocodile attacks. 

### 3.6. GLM Analysis

GLMs showed that the number of attacks on humans and number of attacks on livestock significantly impacted the answers to all but three sub-questions of the five main questions (Table 1). The three questions were as follows: how do Nile crocodiles impact your local community’s lives and livelihoods with respect to attacking livestock (Q1.2) and attacking community members (Q1.4)? What are the local community’s perceptions towards eating Nile crocodile meat in your village with regard to boosting business (Q4.5)?

The GLM suggests a clear relationship between the number of crocodile attacks on humans and the perception that crocodiles restrict freedom of movement (Q1): as attacks increase, more people tend to agree with this statement, and fewer people remain neutral. Regarding attacks on livestock, as the number of crocodile attacks increases, more people tend to remain neutral (Q1.1). As the number of crocodile attacks on humans and livestock increases, the likelihood of respondents strongly agreeing that crocodiles destroy fishing equipment also increases (Q1.2). The local communities’ attitudes towards Nile crocodiles show varying responses based on the type of interaction (attacks on humans or livestock). There is a strong perception of danger associated with livestock attacks (Q2.1) and a generally negative view of honour for both human and livestock attacks (Q2.2). Hate and fear responses are more nuanced (Q2.3 and Q2.4), with livestock attacks not significantly increasing strong feelings of hate or fear. The perception of crocodiles as enemies decreases slightly with increased livestock attacks (Q2.5), indicating complex attitudes influenced by different types of interactions with these predators. The use of Nile crocodiles by local communities reflects a nuanced understanding of their role and impact. There is a strong acknowledgment of the danger they pose, particularly in relation to livestock (Q3.1). Perceptions of honour are mixed, with some community members still holding a positive view despite attacks on humans (Q3.2). Feelings of hate and fear are complex, with neutrality playing a significant role (Q3.3 and Q3.4). The perception of crocodiles as enemies is relatively low (Q3.5). Local community perceptions towards eating Nile crocodile meat are influenced by the context of crocodile attacks. There is a belief in longevity benefits associated with eating crocodile meat, particularly linked to human attacks (Q4.1). The perception of enhanced sexual performance is significant in relation to livestock attacks (Q4.2). Anti-witchcraft beliefs (Q4.3) and medicinal purposes (Q4.4) show a decline in agreement with increased human and livestock attacks, indicating a shift in traditional views possibly due to the negative impact of crocodile conflicts. These perceptions highlight the complex relationship between cultural beliefs and the practical realities of living with crocodiles. The preferred strategies for minimising or mitigating the risks of Nile crocodile attacks vary within the local community (Q5). Establishing crocodile sanctuaries (Q5.1) and migrating to safer areas (Q5.4) are favoured for livestock and human protection, respectively. There is a significant agreement on reducing human–livestock activity (Q5.2) and mixed opinions on the destruction of crocodile breeding habitats (Q5.3). Promoting illegal hunting receives both strong agreement and strong disagreement, reflecting a community divided on this contentious issue. 

## 4. Discussion

We investigated attitudes and perceptions towards crocodiles by applying structured questionnaires. The design of interview questions in structured interviews may bias answers or limit the range of response [31]. Although our survey questions appear biased towards measurements of conflict, we believe, based on our experience working with fisherfolk in the area, that the question reflects fisherfolk attitudes. Despite the emphasis on conflict, several results indicate nuances and complex attitudes. For example, the perception of crocodiles as enemies is relatively low (when asking how crocodiles are used. Our study focuses on conflict in the context of human and livestock mortality by crocodiles [8] but does not address how common or uncommon such conflicts are. Whilst focussing on the conflicts, we also probed positive attitudes to crocodile conservation. It became clear that local communities were not strictly against crocodiles but also sought their conservation through the establishment of sanctuaries. We recommend a follow-up study addressing this specific question, e.g., questioning how often interviewees see crocodiles without conflict. This is important to fully understand the crocodile–human conflict in the current global climate where conflicts—such as crocodile and alligator attacks in Florida or brown bear conflicts in national parks and residential areas—are often sensationalised and predators are villainised for exhibiting natural behaviour where humans encroach into predator habitats, for example, by ever-increasing urbanisation.

Fisherfolk in the Sudd area living near the River Nile perceived that their lives and livelihoods are seriously affected by Nile crocodiles through attacking humans and livestock, not only by the loss of human lives and livestock but also restricting the freedom of movement of communities along the river bank and causing damage to fishing equipment, which is the main operative expenditure of the fisherfolk [4,32]. Damaged fishing nets can be fixed, but it takes time and money [4,33]. While the fishing nets are being repaired, the fisherfolk cannot fish unless they purchase or hire fishing nets. For the fisherfolk to mitigate the impact of crocodile damage on fishing equipment, fisherfolk need to equip themselves with more than eight fishing nets annually, but the extra expense is economically challenging considering the overall poverty. The damage to fishing nets occurs in the flooding season when most fish species migrate to the floodplain for breeding and the crocodiles migrate to the floodplain to feed on these fish species. Similarly, fisherfolk in the Okavango area of Botswana view crocodiles as a serious threat to their lives and equipment, and they delay entrance into the floodplain when water levels drop due to the risk of crocodile attacks [34]. Moreover, crocodiles eat fish from nets. In the dry season between November and January, most crocodiles migrate to breed in the sandy area along the river bank. The reproductive season is considered dangerous as reported by crocodiles in South and Central America [35,36]. During this time, parents strongly warn children not to play in the sandy area along the river bank, and women and girls are advised not to fetch water in dangerous areas. These general views of the various interviewees are set against the backdrop of 23 people being killed and 355 livestock attacked by crocodiles in the same villages between 2018 and 2020 [25]. The GLMs show the number of attacks on both humans and livestock significantly influenced several answers made by the interviewees. Although the attitude of local communities towards the conservation of crocodiles is negative because of the above-mentioned reasons, crocodiles are also actively used by people, and the answers to some of the questions revealed complex and nuanced patterns. Local communities were promoting the eating of Nile crocodile meat to enhance longevity, drug, and sexual potency. On the other hand, certain families and clans have a longstanding tradition of revering crocodiles for cultural, ritual, and spiritual purposes, similar to practices observed in other parts of Africa [37,38,39,40] and elsewhere [41]. The key informant interviews conducted in the Sudd wetlands revealed the presence of over 34 such families, with 14 located in the Southern Zone between Mangalla, Gemeiza, and Terekeka.

Conservation of the Nile crocodile populations and the effective management of their habitats depend on understanding these human attitudes and our ability to recognise and predict species–habitat interactions [42]. People are known to destroy any nesting they come across while collecting resources in the floodplains [34]. Human activities in the immediate surroundings of the wetlands cause significant stress for female crocodiles and can lead to the abandonment of nesting sites and breeding habitats. The rapid population growth demands more resources, and this has a strong link with the expansion of human settlement in a detrimental way to wildlife habitats [43]. Thus, cultural attitudes, the fear of crocodiles because of the real danger, and human habitat encroachment will likely increase the pressure on the species in the future given the increasing human population density and the increasing pressure for development after the end of the civil war. However, the issue is complex as promoting crocodile killing and destruction of their habitats were not considered as crucial strategies to reduce the risks of Nile crocodile attacks by the majority of the local communities. 

This complexity is highlighted by the agreement of local communities on the need to destroy Nile crocodile breeding habitats on the one hand and the need to establish crocodile sanctuaries on the other hand. The polarised views on promoting illegal hunting also underscore the intricate challenges in managing human–crocodile conflicts, necessitating nuanced, community-driven strategies. The low perception of crocodiles as enemies suggests that communities may adopt a balanced or integrated approach to coexisting with crocodiles, recognising their significance beyond the immediate threats they pose. Promoting awareness creation and outreach programs to local communities on conservation education are positive strategies to increase awareness of the real risks and crocodile behaviour, thus reducing human–crocodile conflicts. The nuanced attitudes revealed in certain questions provide a valuable foundation for raising awareness and designing targeted promotional campaigns. The creation of Nile crocodile sanctuaries in selected parts of the Sudd wetlands can minimise the risks of illegal hunting by the local communities especially as this option is being positively considered and accepted by local communities. Such sanctuaries might come with economic benefits from ecotourism. 

During interviews with key informants, the issue of promoting crocodile farming for commercial purposes as a source of income and livelihood was raised as a possible sustainable strategy for the management of this species at the local scale. However, the increasing development of Nile crocodile farming has been associated with diseases transmittable to wild crocodiles and humans [44,45,46,47]. For example, crocodile pond water can have high concentrations of *Salmonella* spp. [48,49,50,51,52,53]. Crocodile meat is an important source of *Salmonella* contamination with a human exposure risk, especially during slaughter and dressing operations [53]. Crocodile meat consumption may lead to infections with a variety of bacteria (*Salmonella* spp., *Vibrio* spp.), parasites (*Spirometra*, *Trichinella*, *Gnathostoma*, pentastomids), and intoxications by biotoxins [54]. Amon viruses, there is a zoonotic disease risk through arboviruses such as Rift Valley Fever virus [55]. Contaminated water can play a role in the transmission of the Rift Valley Fever virus. The virus can be present in the blood, tissues, and bodily fluids of infected crocodiles, and when these contaminated fluids come into contact with water sources, there is a risk of transmission to humans and animals that drink or come into contact with the water. Thus, the introduction of crocodile farms, as widespread elsewhere in Africa including neighbouring Ethiopia [56,57], needs careful consideration of the possible negative effects through increased disease risk to humans and wild crocodiles.

## 5. Conclusions

Fisherfolk communities in the southern zone of the Sudd wetlands are known to engage in activities that impact Nile crocodile populations and their habitats. They not only kill crocodiles directly out of the funded fear of often mortal crocodile attacks on humans and livestock, but they also destroy crocodile nesting sites and gather eggs, which can have negative effects on the ecosystem. The fisherfolk claim that Nile crocodiles often damage their fishing nets, leading to retaliatory killing to compensate for these losses. This practice, combined with the increasing demand for crocodile meat, may encourage poachers to hunt crocodiles illegally, surpassing the limits approved by the Ministry of Wildlife Conservation and Tourism.

To address these challenges and support biodiversity conservation, it is proposed to map Nile crocodile nesting sites and hotspots in the Sudd wetlands. This information could inform government efforts to protect crocodile habitats and create crocodile sanctuaries. Additionally, there is a need to understand and promote the cultural and ritual practices of families and clans in the area, as these may contribute to biodiversity conservation.

A suggested approach to reduce pressure on wild crocodile populations is to create a Nile crocodile sanctuary and promote crocodile farming as a sustainable alternative. Conservation education and outreach programs targeting local communities are also crucial. These programs would aim to mitigate conflicts between humans and crocodiles and raise awareness about the Man and Biosphere Programme, fostering harmonious coexistence with wildlife in the region.

## Figures and Tables

**Figure 2 animals-14-01819-f002:**
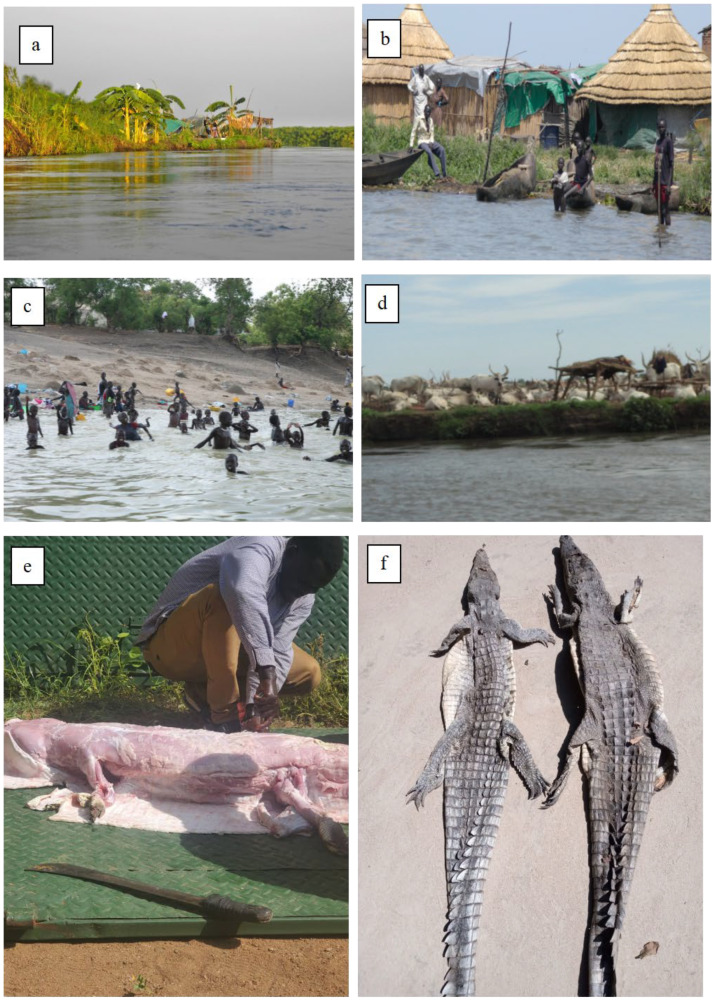
Characteristics of the study area. (**a**) Habitat of Nile crocodiles along the Sudd wetlands, (**b**) human settlement closed to Nile crocodile habitats, (**c**) local community taking risks of bathing in an area near breeding grounds, (**d**) pastoralist communities taking risks of keeping their livestock on the river bank, (**e**) illegal hunting as a revenge by the local communities, and (**f**) dried Nile crocodile skin. Photos by J.S.Benansio.

**Figure 3 animals-14-01819-f003:**
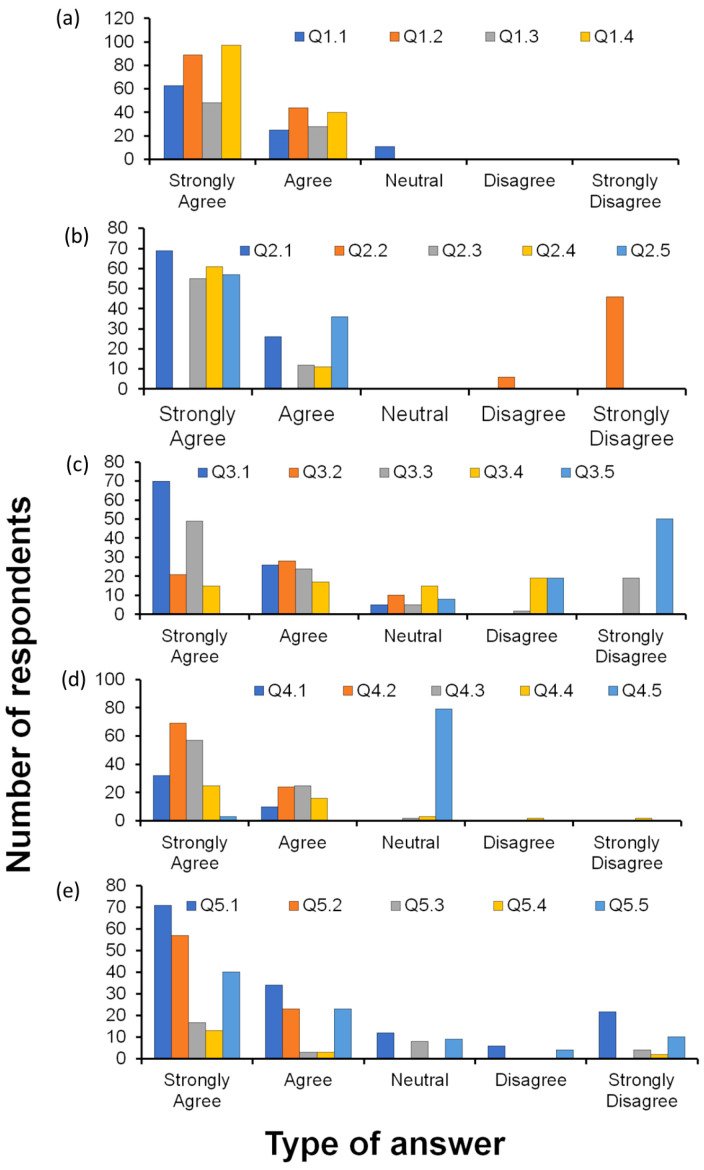
Number of respondents of the answers of the respondents. Data from the different study areas are pooled. Each graphic represents a different question regarding four to five sub-questions: (**a**) How do Nile crocodiles affect the lives and livelihoods of your local community? (**b**) What is the attitude of local communities in your area towards Nile crocodiles? (**c**) How are Nile crocodiles being used by the local community in your village? (**d**) What are the perceptions of local communities towards eating Nile crocodile meat at your village? (**e**) What strategies are preferred by your local community to minimise/mitigate the risks of Nile crocodile attacks on humans and livestock?

**Table 1 animals-14-01819-t001:** GLM results of the distribution of interviewees’ answers to the sub-questions of five main questions. Distributions are shown in Figure 3. Agreement options are listed only for significant results (*p* < 0.05). Predictor variables are the numbers of attacks on humans and attacks on livestock. Listed are the GLM estimates ±SE, the Wald statistics, and *p* values.

	Agreement Option	Predictor: Attacks on	Estimate	±SE	Wald	*p*
How do Nile crocodiles affect the lives and livelihoods of your local community regarding…						
… restricting freedom of movement (Q1.1)?	Agree	humans	0.375	0.098	14.618	0.0001
	Neutral	humans	−0.789	0.317	6.209	0.0127
	Neutral	livestock	0.154	0.0426	13.135	0.0003
… destroying fishing equipment (Q1.3)?	Strongly Agree	humans	0.426	0.209	4.140	0.0419
	Agree	livestock	0.086	0.036	5.556	0.0184
What is the attitude of local communities in your area towards Nile crocodiles regarding…						
… danger (Q2.1)?	Strongly Agree	livestock	0.137	0.050	7.554	0.006
… honour (Q2.2)?	Disagree	humans	0.453	0.169	7.171	0.007
	Strongly Disagree	livestock	0.082	0.0328	6.347	0.012
… hate (Q2.3)?	Strongly Agree	livestock	−0.109	0.039	7.652	0.006
… fear (Q2.4)?	Strongly Agree	livestock	−0.001	0.004	6.571	0.010
	Agree	livestock	−0.059	0.0269	4.813	0.028
… enemy (Q2.5)?	Agree	livestock	−0.073	0.007	107.987	0.000
How are Nile crocodiles being used by the local communities in your village regarding …						
… danger (Q3.1)?	Strongly Agree	livestock	0.151	0.060	6.314	0.012
	Agree	humans	0.290	0.147	3.898	0.048
… honour (Q3.2)?	Strongly Agree	humans	−0.699	0.129	29.180	0.000
	Agree	humans	0.980	0.092	112.479	0.000
… hate (Q3.3)?	Neutral	humans	−0.274	0.084	10.499	0.001
… fear (Q3.4)?	Neutral	humans	0.593	0.200	8.801	0.003
… enemy (Q3.5)?	Disagree	humans	0.399	0.087	20.697	0.000
What are the perceptions of local communities towards eating Nile crocodile meat in your village regarding…						
… longevity (Q4.1)?	Agree	humans	0.239	0.088	7.339	0.006
… enhancing sexually performance (Q4.2)?	Strongly Agree	livestock	0.143	0.044	10.447	0.001
… anti-witchcraft (Q4.3)?	Agree	humans	−0.514	0.067	58.387	0.000
… medicinal purposes (Q4.4)?	Strongly Agree	humans	−0.661	0.123	28.528	0.000
	Agree	livestock	−0.065	0.016	15.554	0.000
	Neutral	livestock	0.040	0.019	4.499	0.033
What strategies are preferred to be used by your local community to minimise/mitigate the risks of Nile crocodile attacks on humans and livestock regarding…						
… establishing crocodile sanctuary (Q5.1)?	Strongly Agree	livestock	0.116	0.054	4.470	0.034
	Neutral	livestock	0.087	0.025	11.430	0.001
… reducing human–livestock activity (Q5.2)?	Agree	humans	0.501	0.190	6.933	0.008
	Neutral	livestock	−0.058	0.025	5.208	0.022
… destruction of crocodile breeding habitats (Q5.3)?	Strongly Agree	humans	−0.295	0.039	56.245	0.000
	Strongly Disagree	humans	0.272	0.120	5.128	0.023
… community migration to safety areas (Q5.4)?	Strongly Agree	humans	−0.468	0.047	98.959	0.000
	Agree	livestock	−0.030	0.006	23.820	0.000
… promoting illegal hunting of crocodiles (Q5.5)?	Strongly Agree	humans	0.767	0.339	5.101	0.023
	Neutral	humans	0.540	0.098	30.104	0.000
	Neutral	livestock	−0.035	0.013	6.983	0.008
	Strongly Disagree	humans	0.763	0.263	8.406	0.003

## Data Availability

Data is contained within the article.

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
