# Peer review of "Attitudes and Perceptions of Local Communities towards Nile Crocodiles (Crocodylus niloticus) in the Sudd Wetlands, South Sudan"

_animals, 2024, doi:10.3390/ani14121819_

Round 1

Reviewer 1 Report

Comments and Suggestions for Authors

The authors conducted interviews with 378 community members in South Sudan to learn about their attitudes about Nile crocodiles with the purpose of providing information that could be used to mitigate conflicts between people and crocodiles and support biodiversity conservation. They conclude that there is in general a negative attitude towards crocodiles, and that the attitudes are complex.  I think the data collected has potential to inform both community members on potential changes in behavior and broader scale regional conservation plans for crocodiles.

That said, I have some concerns about that data collection and analysis.  In the methods states that the questions were open ended, yet then it states that the responses had to be on a Likert scale. The form of the questions in the text are not structured in a way that would lead to valid Likert responses. In addition, I’m not sure that the GLM analysis is appropriate for the way the study was described.

I also feel there are places in the manuscript where the authors could shorten sections by being more direct and including less details that are not as closely related to the study.

Below are some additional comments referenced by line number

Line 40-elsewhere the term fishers is used. Suggest checking for consistency.

Line 72-73- why does a poorly scientifically explored area make it ideal? Is what you mean is it that the need for this information is high?

Line 102- is it the attitudes that can be used or the knowledge of the attitudes that can be used?

Line 131- I’m a little confused about the study sites (5), communities (28 in text, 21 in Appendix), fishing camps (85 line 163).  It would be helpful to see the relationship between the fishing camps and the communities and study sites.  Was the random sample of fishing camps using all of the five study areas or was there an effort to sample equally in the five study sites?  Or proportionally based on the number of fishing camps in each study site? Are the districts the study sites?  Make sure you are consistent in terminology for everything throughout the text.

Line 167- 21 women and 357 men doesn’t really sound like a random sample unless the population(s) are highly male skewed.  Was this really random?  If so, based on what?  Census of the villages?  Was there a minimum age used for selecting the sample?  Looks like it was 19- state those details in the methods.  Were interviews conducted individually?

Line 195 – above on Line 172 it states the questions were open ended which to me means no fixed responses- more recording what the person says and then going back and figuring out the key points.  Line 195 says responses had to be on a Likert scale.  Unless questions were rephrased differently than described in the text I’m not sure they are appropriately worded to get reliable Likert responses.

Line 198- were these discussions done after all of the interviews were conducted?  How was that information translated into information for use in the analysis?

Line 189-203- where is this information used in the analysis?  As above, how is it translated into information used in the quantitative analysis?

Line 215- as I stated above, I’m not sure this is the right analysis for this.

Author Response

Reply to reviewer 1

I have some concerns about that data collection and analysis.  In the methods states that the questions were open ended, yet then it states that the responses had to be on a Likert scale. The form of the questions in the text are not structured in a way that would lead to valid Likert responses.

Reply regarding open-ended: You are correct, the interviews were structured. The actual questions were not open-ended. After asking the questions, we discussed any arising issues in an open-ended manner to assess the general background of attitudes to crocodiles and to biodiversity in general. We now removed “open-ended” from “five open-ended questions”. We added the sentence: “Following these questions, we provided the interviewees with an opportunity to express their attitudes towards crocodiles and biodiversity conservation, aiming to gather general background information.”

Reply regarding Likert responses: Whilst we did not mention Likert responses, the answers had to be (a) strongly agree, (b) agree, (c) neutral, (d) disagree, or (e) strongly disagree, which corresponds to a Likert scale. The written questions appear not to be structured to solicit Likert answers, but they were verbally rephrased with more explanations. E.g., we asked, in the local language: “What is the attitude of local communities in your area towards Nile crocodiles? Do you agree that they pose a danger? Do you agree that they constitute an honour for you? Do you hate them? Do you fear them?  Do you regard them as your enemy?” For the sake of conciseness, we abbreviate this in the text as “What is the attitude of local communities in your area towards Nile crocodiles regarding (Q2.1) danger, (Q2.2) honour, (Q2.3) hate, (Q2.4) fear, and (Q2.5) enemy?”. We modified the text accordingly.

I’m not sure that the GLM analysis is appropriate for the way the study was described.

Reply: Using Generalized Linear Models (GLM) for analysing Likert scale data is a valid approach in many research contexts. GLMs can handle the ordinal nature of Likert scale responses effectively. Likert scale responses are ordinal data, meaning that they have a natural order or ranking. GLMs can handle ordinal response variables by using appropriate link functions, such as the logit or probit link for ordered categorical data. GLMs do not assume that the response variable has a normal distribution, which is often violated in Likert scale data. Instead, GLMs can accommodate different distributions, such as binomial, Poisson, or multinomial, depending on the nature of the response variable. The values of the estimation parameters in the generalized linear model are obtained by maximum likelihood (ML) estimation, which requires iterative computational procedures. Tests for the significance of the effects in the model can be performed via the Wald statistic. Detailed descriptions of these tests can be found in McCullagh and Nelder (1989).

We opted to use GLM for this analysis to test, in a synthetic way, the effect of crocodile attacks on the "lives" and "attitudes" of the population through their answers to the questionnaires. We believe that the results (through the interpretation of the estimate results) give clear indications on the influence of crocodile attacks on population attitudes better, in our opinion, than using other analytical methodologies. In our models, answers were used as dependent variables and the numbers of crocodile attacks on humans and livestock were used as predictors. The estimates of the models indicated whether the probability of a given answer increased/decreased with the increasing number of crocodile attacks.

We expanded the Materials & Methods section accordingly.

I also feel there are places in the manuscript where the authors could shorten sections by being more direct and including less details that are not as closely related to the study.

Reply: We disagree with the reviewer as we feel that all the given information is relevant.

Line 40-elsewhere the term fishers is used. Suggest checking for consistency.

Reply: We now consistently use “fisherfolk”

Line 72-73- why does a poorly scientifically explored area make it ideal? Is what you mean is it that the need for this information is high?

Reply: Yes. We added “…, thus data-deficient …”

Line 102- is it the attitudes that can be used or the knowledge of the attitudes that can be used?

Reply: The latter. Implemented.

Line 131- I’m a little confused about the study sites (5), communities (28 in text, 21 in Appendix), fishing camps (85 line 163).  It would be helpful to see the relationship between the fishing camps and the communities and study sites.  Was the random sample of fishing camps using all of the five study areas or was there an effort to sample equally in the five study sites?  Or proportionally based on the number of fishing camps in each study site? Are the districts the study sites?  Make sure you are consistent in terminology for everything throughout the text.

Reply: The confusion stems from our use of the word “communities”, which we applied as equivalent to villages. We corrected the first sentence of the paragraph accordingly: “We studied 21 different villages (Appendix A) and their 85 associated fishing camps from the following five administrative areas of the Southern Zone of Sudd Wetlands”. In the subsequent paragraph, we added: “The random selection was across the five administrative areas without a proportional representation of the administrative areas.” Yes, the five study sites are equivalent to the administrative areas.

Line 167- 21 women and 357 men doesn’t really sound like a random sample unless the population(s) are highly male skewed.  Was this really random?  If so, based on what?  Census of the villages?  Was there a minimum age used for selecting the sample?  Looks like it was 19- state those details in the methods.  Were interviews conducted individually?

Reply: Women were underrepresented. Participant selection was random by approaching potential participants randomly on sight whilst interviewers visited the fishing camps. We only interviewed adults aged 19 or above. Interviews were conducted individually. We added all the info to the text.

Line 195 – above on Line 172 it states the questions were open ended which to me means no fixed responses- more recording what the person says and then going back and figuring out the key points.  Line 195 says responses had to be on a Likert scale.  Unless questions were rephrased differently than described in the text I’m not sure they are appropriately worded to get reliable Likert responses.

See our reply regarding Likert responses above.

Line 198- were these discussions done after all of the interviews were conducted?  How was that information translated into information for use in the analysis?

Reply: Yes. See below

Line 189-203- where is this information used in the analysis?  As above, how is it translated into information used in the quantitative analysis?

Reply: It was not quantitatively analysed but used for us to interpret the results of the questionnaires. We added: “All this information guided us in interpreting the collected quantitative data.”

Line 215- as I stated above, I’m not sure this is the right analysis for this.

See our reply regarding Likert responses above.

Reviewer 2 Report

Comments and Suggestions for Authors

Summary:

The aim of this study is to understand the perceptions and attitudes of humans towards crocodiles in South Sudan. This problem is of great significance because, though the Nile crocodile is considered a species of “least concern,” human-crocodile conflict poses significant threat to both human life and livelihood and conservation of  local crocodile populations. Data were collected via survey interviews from community members throughout South Sudan. Surveys included questions regarding perceptions and attitudes towards crocodiles, impact of human-crocodile conflict on lives and livelihoods, and opinions on human-crocodile conflict mitigation strategies. Survey results were compared to observed human-crocodile conflict events from another study (Benansio et al. 2022) to estimate the influence of conflict events on perceptions and attitudes. Results suggest community members hold strong negative perceptions and attitudes towards crocodiles, though they support crocodile conservation practices for their cultural and economic value via consumption of meat and trade in skin/leather goods. These findings are situated in a discussion on the role of crocodilian management in regard to increasing human densities and development of crocodilian habitats for human habitation that will inevitably increase the risk of human-crocodile conflict in the future.

General Comments:

Thorough and well-formed background on the study’s topic of interest. The importance of understanding human-crocodile conflict for not only improving crocodilian conservation but also protecting human lives and livelihoods is well framed. This issue is particularly important in areas where human-crocodilian interactions cannot be avoided, such as in areas where interaction with crocodilian habitat (e.g., through fishing)  is central to the human economy and livelihood.

Methods are clear, though I do wonder how interviewees may have responded to more open-ended questions or prompts that included potential positive attitudes towards crocodiles. The options in the survey questions appear biased towards measurements of conflict with little opportunity for interviewees to demonstrate any neutral or positive perceptions of crocodiles. For example, how often do these communities simply see crocodiles without conflict occurring (most of the time conflict occurs versus rarely conflict occurs)? This type of information would have been helpful to further explain the degree of conflict in these communities. While they do present data on observed conflict events, these data do not inform readers how common or uncommon conflict events are. I do not raise this point with intention to diminish the significant impact of human-crocodilian conflicts; however, I believe it is important to address the frequency of conflict events in the current global climate where predators are villainized by the media for exhibiting natural behaviors in an ever-increasing urbanized landscape. As scientists we hold a responsibility to share an impartial point-of-view rather than potentially contributing to public fear of wildlife driven by media portrayal of those wildlife. 

Survey result narrative is fairly clear. I believe the GLM results (Section 3.6) warrants greater exposition.

Discussion is sound and includes connections to relevant literature.

Specific Comments:

Abstract: The abstract does not adequately address some areas of the study’s methods - for example, the use of GLMs to assess correlation between observed human-crocodile conflict events from another study and human perceptions and attitudes concerning crocodiles found in this study. The information presented in Lines 211 - 224 should be summarized to increase clarity in the abstract. “GLM” is also not defined appropriately in the abstract. 

Line 142: “support” may be a more appropriate word than “allow” in a population and community ecology context. “Abundant populations of ungulates support the presence of large crocodile populations.” 

Line 170: Define “minors.”

Line 254: Who “stated” in this sentence? Use of “practically” is confusing here as well. I am not sure what this part of the sentence is trying to say, “..use to attack their livestock at night practically in the dry season…”

Line 355: It would improve clarity to clearly state which two sub-questions are deemed to lack significant impact and report the related statistic. - I do believe this whole sub-section (3.6) warrants greater exposition rather than relying on the table alone to support the abstract comment, “GLM indicating a direct link between the number of crocodile attacks and human attitudes,” and Discussion comment, “The GLM models show the number of attacks on both humans and livestock significantly influenced several answers made by the interviewees.” This is a potentially important point for researchers and wildlife managers that should be well explained in the results. 

Table 1:I find Table 1 difficult to read and interpret as currently presented. Is it possible for the authors to present the most important results to a barplot figure and retain the table as an appendix for readers who would like to see more detailed statistics. 
If a figure is not possible, the authors might consider updating the format to more clearly display the results. For example, they might left-justify text and indent subheadings to more clearly demonstrate hierarchical relationships. They could also consider presenting an abbreviated table with only significant results in the manuscript and retain the detailed table as an appendix. 

Comments on the Quality of English Language

Minor grammatical errors throughout: Line 253-255, Line 389-390, Line 392-393. Note this is not an exhaustive list as I focused my review on scientific contributions rather than editorial comments as instructed.

The abstract includes several grammatical errors (e.g., sentence fragments [Line 47-48], undefined pronouns [Line 45-47], missing punctuation [Line 48-49]) that reduce readability. 

Author Response

Reply to reviewer 2

Methods are clear, though I do wonder how interviewees may have responded to more open-ended questions or prompts that included potential positive attitudes towards crocodiles. The options in the survey questions appear biased towards measurements of conflict with little opportunity for interviewees to demonstrate any neutral or positive perceptions of crocodiles. For example, how often do these communities simply see crocodiles without conflict occurring (most of the time conflict occurs versus rarely conflict occurs)? This type of information would have been helpful to further explain the degree of conflict in these communities. While they do present data on observed conflict events, these data do not inform readers how common or uncommon conflict events are. I do not raise this point with intention to diminish the significant impact of human-crocodilian conflicts; however, I believe it is important to address the frequency of conflict events in the current global climate where predators are villainized by the media for exhibiting natural behaviors in an ever-increasing urbanized landscape. As scientists we hold a responsibility to share an impartial point-of-view rather than potentially contributing to public fear of wildlife driven by media portrayal of those wildlife. 

Reply: Excellent point and we agree. We now discuss this caveat in the first paragraph of the discussion.

I believe the GLM results (Section 3.6) warrants greater exposition.

Specific Comments:

Abstract: The abstract does not adequately address some areas of the study’s methods - for example, the use of GLMs to assess correlation between observed human-crocodile conflict events from another study and human perceptions and attitudes concerning crocodiles found in this study. The information presented in Lines 211 - 224 should be summarized to increase clarity in the abstract. “GLM” is also not defined appropriately in the abstract. 

Reply: Implemented. “To assess whether responses were influenced by the intensity of crocodile threats, published data on fatal crocodile attacks on humans and livestock were analysed using Generalized Linear Models (GLM). This analysis indicated a direct link between the number of crocodile attacks and human attitudes.”

Line 142: “support” may be a more appropriate word than “allow” in a population and community ecology context. “Abundant populations of ungulates support the presence of large crocodile populations.” 

Reply: Implemented.

Line 170: Define “minors.”

Reply: Done.

Line 254: Who “stated” in this sentence? Use of “practically” is confusing here as well. I am not sure what this part of the sentence is trying to say, “..use to attack their livestock at night practically in the dry season…”

Reply: Implemented. “For instance, focus group discussions highlighted that Nile crocodile attack livestock at night mainly in the dry season when there are no more fish in the floodplain.”

Line 355: It would improve clarity to clearly state which two sub-questions are deemed to lack significant impact and report the related statistic. - I do believe this whole sub-section (3.6) warrants greater exposition rather than relying on the table alone to support the abstract comment, “GLM indicating a direct link between the number of crocodile attacks and human attitudes,” and Discussion comment, “The GLM models show the number of attacks on both humans and livestock significantly influenced several answers made by the interviewees.” This is a potentially important point for researchers and wildlife managers that should be well explained in the results. 

Reply: Implemented by expanding section 3.6 and emphasising the complex and nuanced patterns revealed by some of the questions.

Table 1: I find Table 1 difficult to read and interpret as currently presented. Is it possible for the authors to present the most important results to a barplot figure and retain the table as an appendix for readers who would like to see more detailed statistics. 
If a figure is not possible, the authors might consider updating the format to more clearly display the results. For example, they might left-justify text and indent subheadings to more clearly demonstrate hierarchical relationships. They could also consider presenting an abbreviated table with only significant results in the manuscript and retain the detailed table as an appendix. 

 Reply: There seems no adequate graphic method to show the results. We simplified the table as suggested by retaining only significant results and by reformatting.

Comments on the Quality of English Language

 Reply: One of the co-authors (JEF), who is native English speaker, revised the manuscript.